# The visual speech head start improves perception and reduces superior temporal cortex responses to auditory speech

Patrick J Karas, John F Magnotti, Brian A Metzger, Lin L Zhu, Kristen B Smith, Daniel Yoshor, Michael S Beauchamp*

Department of Neurosurgery, Baylor College of Medicine, Houston, United States

**Abstract** Visual information about speech content from the talker's mouth is often available before auditory information from the talker's voice. Here we examined perceptual and neural responses to words with and without this visual head start. For both types of words, perception was enhanced by viewing the talker's face, but the enhancement was significantly greater for words with a head start. Neural responses were measured from electrodes implanted over auditory association cortex in the posterior superior temporal gyrus (pSTG) of epileptic patients. The presence of visual speech suppressed responses to auditory speech, more so for words with a visual head start. We suggest that the head start inhibits representations of incompatible auditory phonemes, increasing perceptual accuracy and decreasing total neural responses. Together with previous work showing visual cortex modulation (Ozker et al., 2018b) these results from pSTG demonstrate that multisensory interactions are a powerful modulator of activity throughout the speech perception network.
DOI: https://doi.org/10.7554/eLife.48116.001

*For correspondence:
michael.beauchamp@bcm.edu

Competing interests: The authors declare that no competing interests exist.

## Introduction

Pairing noisy auditory speech with a video of the talker dramatically improves perception (*Bernstein et al., 2004*; *Grant and Seitz, 2000*; *Munhall et al., 2004*; *Ross et al., 2007*; *Sumby and Pollack, 1954*). Visual speech can provide a perceptual benefit because any particular mouth movement made by the talker is compatible with only a few auditory phonemes (*Cappelletta and Harte, 2012*; *Jeffers and Barley, 1971*; *Neti et al., 2000*). For instance, viewing a rounded open-mouth shape allows the observer to rule out the ~80% of phonemes incompatible with this mouth shape (*Figure 1A*). Bayesian inference provides a plausible algorithm for how the observer combines the visual mouth shape and auditory phoneme information (*Ma et al., 2009*; *Magnotti and Beauchamp, 2017*).

A second potential reason for the perceptual benefit of visual speech has received less attention (*Peelle and Sommers, 2015*): visual speech can provide a head start on perception when mouth movements begin before auditory vocalization. This extra processing time is important because humans must understand speech both quickly (as speech is generated rapidly, at rates of ~5 syllables a second) and accurately (as errors in communication are potentially costly). In order to examine the perceptual and neural consequences of the visual speech head start, we leveraged the natural variability in the temporal relationship between the auditory and visual modalities (*Chandrasekaran et al., 2009*; *Schwartz and Savariaux, 2014*). Although there is a range in the relative onset of auditory and visual speech, most audiovisual words provide a visual head start. We term these words 'mouth-leading', as the visual information provided by the talker's mouth is available before auditory information from the talker's voice. For instance, in an audiovisual recording of the word 'drive' (*Figure 1B*) the visual onset of the open mouth required to enunciate the initial 'd'

of the word preceded auditory vocalization by 400 ms, allowing the observer to rule out incompatible auditory phonemes (and rule in compatible phonemes) well before any auditory speech information is available. We term words that do not provide a visual head start as 'voice-leading:' auditory information from the talker's voice is available before visual information provided by the talker's mouth. For instance, in an audiovisual recording of the word 'known', auditory voice onset occurred 100 ms before visible changes in the talker's mouth (*Figure 1C*). If the visual head start contributes to the perceptual benefits of visual speech, these benefits should be greater for mouth-leading words than for voice-leading words.

Any influence of the visual head start on speech perception should be reflected in measurements of the activity of the neural circuits important for speech perception. Studies of patients with cortical lesions (*Hickok et al., 2018*; *Stasenko et al., 2015*) and fMRI, MEG, EEG, and electrocorticographic (intracranial EEG) studies have implicated a network of brain areas in occipital, temporal, frontal, and parietal cortex (*Crosse et al., 2016*; *Hickok and Poeppel, 2015*; *Okada et al., 2013*; *Salmelin, 2007*; *Shahin et al., 2018*; *Sohoglu and Davis, 2016*; *van Wassenhove et al., 2005*). Within this network, posterior superior temporal gyrus and sulcus (pSTG) are responsive to both unisensory auditory speech (*Belin et al., 2000*; *Formisano et al., 2008*; *Mesgarani et al., 2014*) and unisensory visual speech (*Bernstein et al., 2011*; *Bernstein and Liebenthal, 2014*), with subregions responsive to both auditory and visual speech (*Beauchamp et al., 2004*; *Ozker et al., 2017*; *Ozker et al., 2018a*; *Rennig et al., 2018*; *Zhu and Beauchamp, 2017*). We examined the neural differences between the processing of mouth-leading and voice-leading speech in pSTG using intracranial EEG. This technique has the advantage of both high spatial resolution, necessary to measure activity from focal areas within the pSTG, and high temporal resolution, required to capture the small auditory-visual asynchrony differences between mouth-leading and voice-leading words.

## Results

### Perceptual results

In the perceptual experiments, participants identified auditory-only and audiovisual words with and without added auditory noise. Perception was very accurate for words without added auditory noise. Adding noise reduced perceptual accuracy below ceiling, revealing differences between conditions. To evaluate these differences, we constructed a generalized linear mixed-effects model (GLMM) with fixed effects of word format (noisy auditory-only *vs.* noisy audiovisual) and word type (mouth-leading *vs.* voice-leading) and their interaction. All models included random effects for participant; exemplar was included as a random effect for the second experiment because of the larger number of stimuli.

In the first perceptual experiment (*Figure 2A*), 40 participants were presented with sixteen different stimuli consisting of four stimulus exemplars (two mouth-leading words and two voice-leading words) in each of the four formats (clear auditory, noisy auditory, clear audiovisual, noisy audiovisual). For mouth-leading words, viewing the face of the talker increased the intelligibility of noisy auditory speech by 23%, from 3% for auditory-only words to 26% for audiovisual words (odds-ratio 0.00 for auditory *vs.* 0.07 for audiovisual; $p<10^{-16}$). For voice-leading words, viewing the face of the talker provided only a 10% accuracy increase, from 14% to 24% (odds-ratio 0.03 *vs.* 0.09, $p=10^{-9}$). The interaction between format and word type was significant ($p=10^{-14}$). In addition to the interaction, there were significant main effects of format ($p<10^{-16}$) and word type ($p=10^{-15}$) driven by higher accuracy for audiovisual words and for voice-leading words.

To extend these results to a larger stimulus set, in the second perceptual experiment (*Figure 2B*), 46 participants were presented with forty stimuli (different than those used in the first perceptual experiment) consisting of ten stimulus exemplars (five mouth-leading words and five voice-leading words) presented in each of the four formats. For these mouth-leading words, viewing the face of the talker increased the intelligibility of noisy auditory speech by 59%, from 6% for auditory-only words to 65% for audiovisual words (odds-ratio 0.01 *vs.* 2.24, $p<10^{-16}$). For voice-leading words, viewing the face of the talker provided only a 37% accuracy increase, from 23% to 60% (odds-ratio 0.17 *vs.* 1.66, $p<10^{-16}$). The interaction between format and word type was significant ($p<10^{-16}$)

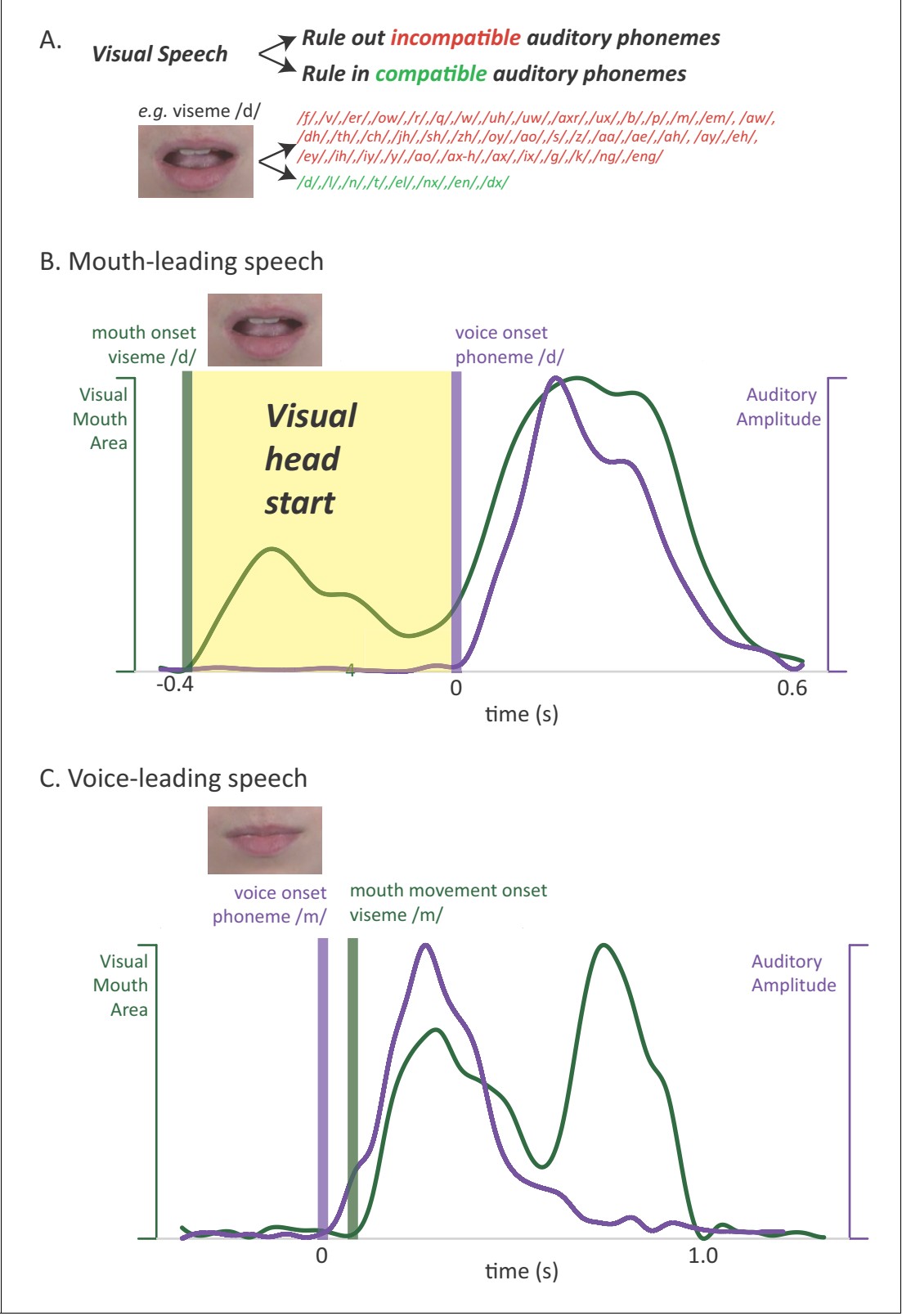

**Figure 1.** Relationship between auditory and visual speech. (**A**) Visual speech provides independent information about speech content. Any given visual speech feature, such as the open mouth visible when pronouncing 'd', is incompatible with many auditory phonemes (red) and compatible with a few auditory phonemes (green). (**B**) Visual speech provides a processing head start on auditory speech (yellow region) as shown by auditory and visual speech feature asynchrony for the word 'drive.' Audiovisual speech is composed of visual mouth movements (green line showing visual mouth area) and

*Figure 1 continued on next page*

*Figure 1 continued*

auditory speech sounds (purple line showing auditory sound pressure level). Lip and mouth movements (visual speech onset, green bar) occur prior to vocalization (auditory speech onset, purple bar). Time zero is the auditory speech onset. This word is classified as 'mouth-leading' as visual mouth movements begin before auditory speech. (C) For the word 'known,' mouth movements begin after auditory vocalization (green bar comes after purple bar) and there is no visual head start. This word is termed as 'voice-leading' because vocalization begins before visible mouth movements.

DOI: https://doi.org/10.7554/eLife.48116.002

driven by the larger benefit of visual speech for mouth-leading words. In addition to the interaction, there was a significant main effect of format ($p<10^{-16}$) and word type (p=0.04) driven by higher accuracy for audiovisual words and for voice-leading words.

## Neural results

The perceptual experiments suggest that the visual head start is an important contributor to the benefits of visual speech. To study the neural differences between words with and without a visual head start, in eight epileptic participants we recorded from electrodes implanted bilaterally over the posterior superior temporal gyrus (pSTG) that showed a significant response to auditory-only speech measured as the percent increase in the power of the high-frequency (75 to 150 Hz) electrical activity relative to baseline (*n* = 28 electrodes, locations and auditory-only response magnitude shown in *Figure 3A*). In contrast to the perceptual studies, where both clear and noisy speech was presented, in the neural experiments only clear speech was presented in order to maximize the size of the neural response. The stimulus exemplars consisted of the two mouth-leading words and the two voice-leading words used in the first perceptual experiment presented in auditory-only, visual-only, and audiovisual formats, resulting in twelve total stimuli.

As shown in *Figure 3B*, the neural response to the single word stimuli in the pSTG began shortly after auditory speech onset at 0 ms, peaked at 180 ms, and returned to baseline after auditory speech offset at 550 ms. To quantify the responses, we entered the mean response across the window from auditory speech onset to offset (0 ms to 550 ms) relative to baseline into a linear mixed-effects model (LME) with fixed effects of word format (auditory-only *vs.* audiovisual), word type (mouth-leading *vs.* voice-leading) and the interaction.

As shown in *Figure 3C*, the response to audiovisual mouth-leading words was 34% smaller than the response to auditory-only mouth-leading words (118% *vs.* 152%, $p=10^{-5}$). In contrast, the response to audiovisual voice-leading words was not significantly different than the response to auditory-only voice-leading words (129% *vs.* 134%, p=0.4). This interaction between word format and word type was significant (p=0.003). There were also significant main effects of word format ($p=10^{-6}$, driven by larger responses for auditory-only words) and word type (p=0.01, driven by larger responses for mouth-leading words).

## Multisensory influence of visual speech: Mouth-Leading words

For the visual speech head start to influence processing in pSTG, the visual speech information must be present in pSTG and it must arrive early enough to exert multisensory influence. To determine if this was the case, we examined the responses to visual-only words. There was a significant positive response to visual-only words, with a mean amplitude of 20% (0 to 200 ms after visual speech onset; significantly greater than baseline, $p=10^{-7}$, single sample *t*-test). The effect was consistent across electrodes, with 27 out of 28 electrodes showing a positive response to visual-only speech, demonstrating that information about visual speech reaches pSTG.

The earlier that visual information arrives in the pSTG, the more opportunity it has to influence auditory processing. As shown in *Figure 4A*, the pSTG response to visual speech features occurred ~100 ms earlier than the response to auditory speech features, sufficient time for substantial multisensory interactions to occur. To further quantify this observation, we calculated the latency of the response (defined as the time of half-maximum response) to visual-only and auditory-only speech in individual electrodes. Responses were aligned to auditory onset (or to the time when auditory onset would have occurred for visual-only stimuli), with the response to visual-only speech occurring a mean of 123 ms earlier than the response to auditory-only speech (paired *t*-test,

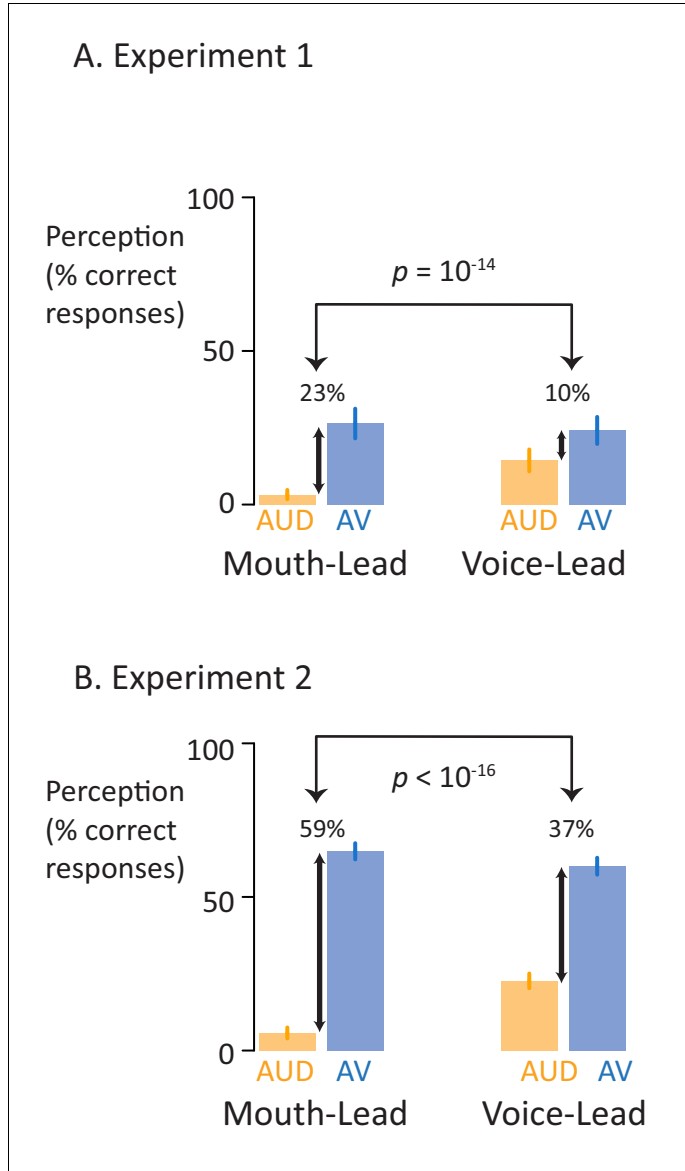

**Figure 2.** Perceptual performance on speech-in-noise recognition tasks. (**A**) Results of the first perceptual experiment on two mouth-leading words and two voice-leading words. For the mouth-leading words (left plot), the addition of visual speech increased comprehension of noisy auditory words by 23% (small black arrow showing difference between noisy auditory-only speech, orange bar labeled AUD, and noisy audiovisual speech, blue bar labeled AV). In contrast, for voice-leading words (right plot) the addition of visual speech increased accuracy by only 10%. The interaction (difference of the differences) was significant ($p=10^{-14}$). Error bars represent standard error of the mean across subjects. (**B**) Results of the second perceptual experiment on five mouth-leading words and five voice-leading words, all different than those used in the first perceptual experiment. Adding visual speech to noisy auditory speech produced a significantly greater enhancement ($p<10^{-16}$) for mouth-leading words (left plot) than for voice-leading words (right plot).

DOI: https://doi.org/10.7554/eLife.48116.003

$p=10^{-8}$). For 25 out of 28 electrodes, the response to visual-only mouth-leading words occurred earlier than the response to auditory-only mouth-leading words.

If visual and auditory speech processing interacts in pSTG, stronger responses to visual speech might result in more powerful multisensory interactions. To test this idea, we calculated the amplitude of the early neural response to visual-only mouth-leading words (0 to 200 ms following the onset of the first visual mouth movement) and compared it with the multisensory effect, defined as

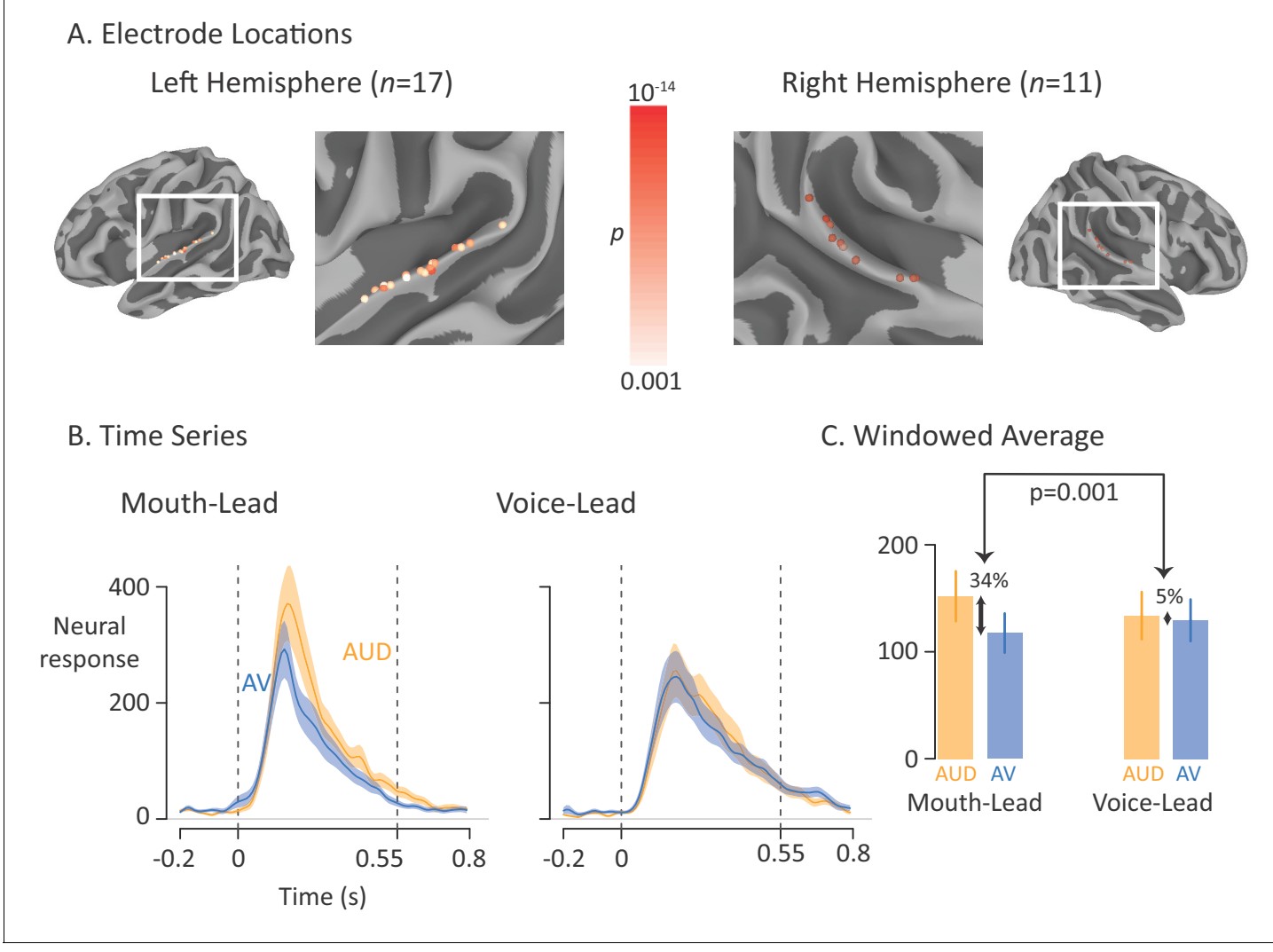

**Figure 3.** Average broadband high-frequency activity by experimental condition. (A) The location of 17 left-hemisphere (left panel) and 11 right-hemisphere (right panel) electrodes that met both an anatomical criterion (located over the posterior superior temporal gyrus) and a functional criterion (significant response to auditory-only speech). The color of each electrode shows the significance (corrected for multiple comparisons using $p<0.001$ Bonferroni-corrected) of each electrode's response to the auditory-only condition during the period from auditory speech onset to offset. (B) For mouth-leading words (left panel), the neural response to auditory-only words (AUD; orange line) was greater than the response to audiovisual words (AV; blue line). For voice-leading words (right panel), the responses were similar. Shaded regions show standard error of the mean across electrodes ($n = 28$) and dashed lines show auditory speech onset (0 s) and offset (0.55 s). (C) To quantify the difference between word types, the neural response was averaged within the window defined by auditory speech onset and offset. For mouth-leading words (left panel), the auditory-only format evoked a significantly greater response than the audiovisual format (34% difference, $p=10^{-5}$). For voice-leading words, there was little difference (5%, $p=0.41$), resulting in a significant interaction between word format and word type (34% vs. 5%, $p=0.001$). Error bars show standard error of the mean across electrodes ($n = 28$).

DOI: https://doi.org/10.7554/eLife.48116.004

The following figure supplement is available for figure 3:

**Figure supplement 1.** Anatomic distribution of multisensory gain by electrode.

DOI: https://doi.org/10.7554/eLife.48116.005

amplitude of the reduction for audiovisual vs. auditory-only mouth-leading words. As shown in *Figure 4B*, there was a significant positive relationship across electrodes between early visual response and multisensory influence ($r = 0.64$, $p=10^{-4}$).

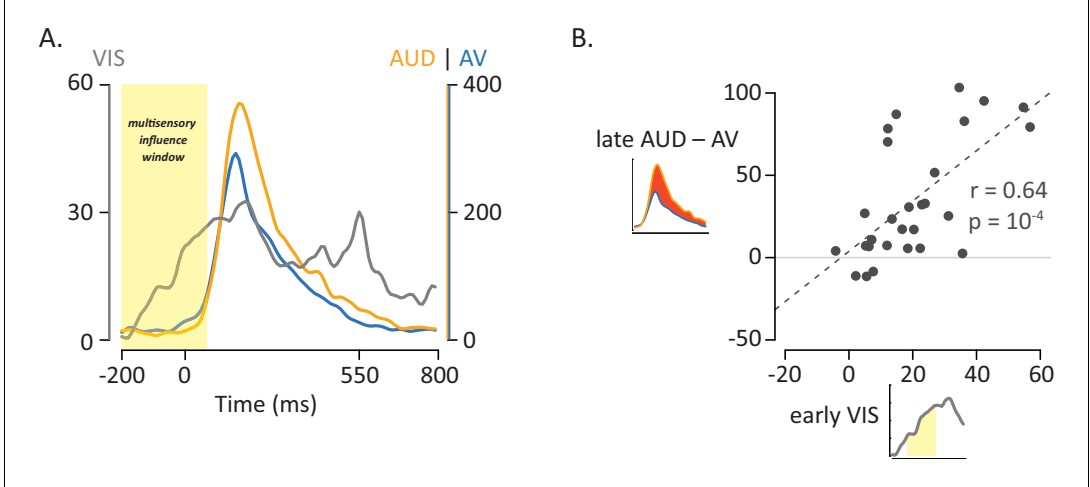

**Figure 4.** Influence of Visual Speech on Multisensory Integration. (**A**) Responses to the three formats of mouth-leading words (visual-only, gray; auditory-only, orange; audiovisual, blue). Responses were aligned to auditory onset at $t = 0$ (or to the time when auditory onset would have occurred for visual-only stimuli). The left vertical axis contains the scale for visual-only neural responses (0% to 60%), which were much smaller than auditory-only and audiovisual responses (scale given in right-hand vertical axis; 0% to 400%). The visual-only response onset occurred prior to auditory-only response onset, creating a multisensory influence window (yellow region) in which visual speech could influence processing of auditory speech, resulting in a reduced response to audiovisual compared with auditory-only speech. (**B**) The amplitude of the early neural response (BHA) to visual-only stimuli was positively correlated with the difference in the neural response between audiovisual and auditory-only speech ($N = 28$; $r = 0.64$, $p=10^{-4}$). The early visual-only response (horizontal axis) for each electrode was the average BHA for the 200 ms period following visual speech onset ($-100$ ms to 100 ms; yellow region in axis inset). The difference between the audiovisual and auditory-only neural response (vertical axis) was calculated as the difference in average BHA during auditory speech (0 ms to 550 ms; red region in axis inset).

DOI: https://doi.org/10.7554/eLife.48116.006

The following figure supplement is available for figure 4:

**Figure supplement 1.** Influence of Visual Speech for Voice-Leading Words.

DOI: https://doi.org/10.7554/eLife.48116.007

## Multisensory influence of visual speech: Voice-Leading words

For voice-leading words, there was no processing head start, as auditory speech began at the same time or even earlier than visual speech. This suggests that there should be less opportunity for multisensory influence, a prediction verified by examining the neural response to voice-leading words: the neural response to voice-leading words presented in the audiovisual and auditory-only formats were similar (129% *vs.* 134%, p=0.4). This could not be explained by a lack of response to the visual component of the words, as there was a robust neural response to visual-only voice-leading words (*Figure 4—figure supplement 1A*). Instead, it likely reflected less opportunity for visual speech to influence auditory processing due to the relative timing of the two modalities. For voice-leading words, the latency of the visual-only and auditory-only neural responses were similar (mean latency 11 ms later for visual-only *vs.* auditory-only speech, p=0.77) reflecting the similar physical onset times of visual and auditory speech. Consistent with the idea that the later arrival of visual speech information lessened multisensory, across electrodes there was not a significant correlation between visual-only response amplitude and multisensory influence, defined as the difference between the audiovisual *vs.* auditory-only neural responses (*Figure 4—figure supplement 1B*; $r = 0.18$, p=0.35).

## Control analysis: Latency

Our analysis depended on accurate time-locking between stimulus presentation and neural response recording. A photodiode placed on the monitor viewed by the participants was used to measure the actual onset time of visual stimulus presentation, while a splitter duplicated the auditory output from the computer to measure the actual onset time of auditory stimulus presentation. Both signals were recorded by the same amplifier used to record neural data, ensuring accurate synchronization. Latencies were similar for auditory-only mouth-leading *vs.* voice-leading words (128 ms *vs.* 134 ms,

p=0.54) demonstrating that the alignment of responses to the physical onset of speech was effective.

## Control analysis: Anatomical specialization

Previous studies have described anatomical specialization within pSTG (*Hamilton et al., 2018*; *Ozker et al., 2017*). Electrodes were color coded by the amplitude of the multisensory influence, calculated as the difference between auditory-only and audiovisual mouth-leading words. When viewed on the cortical surface, no consistent organization of multisensory influence was observed (*Figure 3—figure supplement 1*).

## Discussion

It has long been known that viewing the talker's face enhances the intelligibility of noisy auditory speech (*Sumby and Pollack, 1954*). Visual speech can enhance intelligibility in two related ways (*Peelle and Sommers, 2015*). First, independent information about speech content from the visual and auditory modalities allows for the application of multisensory integration to more accurately estimate speech content. Second, the earlier availability of visual speech information allows for a head start on processing. To examine the contributions of the visual head start to perception, we took advantage of the natural variability between mouth-leading words, in which a visual head start is provided by the onset of visual speech before auditory speech, and voice-leading words, for which there is no head start (*Chandrasekaran et al., 2009*; *Schwartz and Savariaux, 2014*).

The results of our perceptual experiments show that viewing the face of the talker increases intelligibility for both mouth-leading words and voice-leading words, as expected. However, the benefit of visual speech was significantly greater for mouth-leading words, demonstrating that the visual head start enhances perception beyond the mere availability of visual information. Mirroring the perceptual effects, in neural recordings from the pSTG there was a significant difference in the effects of visual speech during presentation of mouth-leading and voice-leading words. Surprisingly, when visual speech was present for mouth-leading words, the neural response *decreased*, the opposite of the *increased* accuracy observed perceptually.

To better understand these results, we constructed a *post hoc* neural model that builds on the experimental observation that the pSTG contains populations selective for specific phonemes (*Formisano et al., 2008*; *Hamilton et al., 2018*; *Mesgarani et al., 2014*). The key feature of the model is that for mouth-leading words (words with a visual head start) the early availability of visual information enhances responses in neurons representing phonemes that are compatible with the visual speech and suppresses neurons representing phonemes that are incompatible with the visual speech. In all published classifications, there are more incompatible than compatible phonemes for any particular viseme (*Cappelletta and Harte, 2012*; *Jeffers and Barley, 1971*; *Neti et al., 2000*). Therefore, the visual head start *decreases* the total response since more neuronal populations are suppressed than are enhanced. This is shown graphically in *Figure 5B* for the auditory-only syllable 'd' and in *Figure 5C* for audiovisual 'd'. For voice-leading speech such as the audiovisual syllable 'm' (*Figure 5D*) there is less time for the suppression and enhancement of incompatible and compatible phonemes to manifest, resulting in similar responses to audiovisual and auditory-only voice-leading words.

The *post hoc* neural model provides a qualitative explanation for the decreased neural response to words with a visual head start. The model also provides a qualitative explanation for the perceptual benefit of the visual head start. The key determinant of perceptual accuracy is the signal-to-noise ratio, defined as the ratio between the response of the neurons selective for the presented phoneme and all other populations. Adding noise to auditory stimuli decreases neuronal responses, making it more difficult to determine the population with the strongest response (low signal-to-noise ratio). The visual head start increases the response of neurons responding to the presented auditory phoneme and decreases the response of most other populations because any given visual speech feature is incompatible with most phonemes. This creates a larger response difference between the two populations (higher signal-to-noise ratio) making perception more accurate relative to auditory-only speech. This process is illustrated schematically for noisy auditory 'da' and noisy audiovisual 'da' in *Figure 5E*.

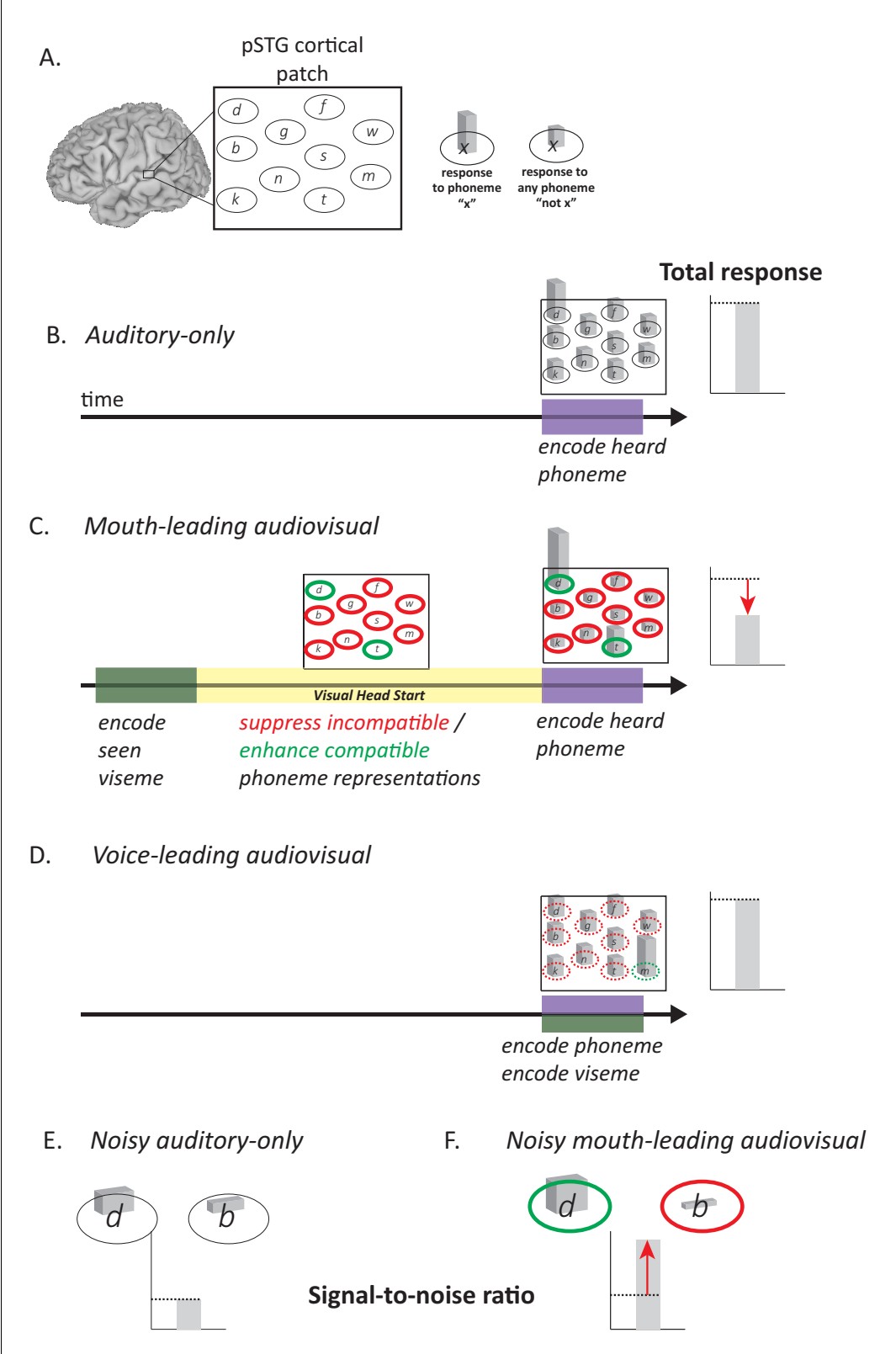

**Figure 5.** Model of Audiovisual Interactions in pSTG. (**A**) In the pSTG, small populations of neurons are selective for specific speech sounds (phonemes). Each population is shown as an ellipse labeled by its preferred phoneme. The ellipses are shown spatially separated but the model is equally applicable if the neurons are intermixed instead of spatially segregated. Selectivity is only partial, so that for the population of neurons selective for a given phoneme 'x' (ellipse containing 'x') presentation of the phoneme 'x' evokes a larger response than presentation of any other phoneme ('not

*Figure 5 continued on next page*

*Figure 5 continued*

x'). (**B**) When an auditory phoneme is presented, all populations of neurons respond, with the highest response in the population of neurons selective for that phoneme. Example shown is for presentation of auditory 'd'; the amplitude of the response in each population of neurons is shown by the height of the bar inside each ellipse, with highest bar for 'd' population. The total response summed across all populations is shown at right. (**C**) For mouth-leading speech, early arriving visual speech provides a head start (yellow region). During this time, activity in neuronal populations representing incompatible phonemes is suppressed (red outlines) and activity in neuronal populations representing compatible phonemes in enhanced (green outlines). Arrival of auditory speech evokes activity in all populations. Because there are more suppressed populations than enhanced populations, the total response across all populations is decreased relative to the auditory-only format (dashed line and red arrow). Example shown is for audiovisual 'd', resulting in larger responses in populations representing the compatible phonemes 'd' and 't', smaller responses in all other populations. (**D**) For voice-leading speech, visual speech and auditory speech onset at similar times, resulting in no opportunity for suppression or enhancement (dashed outlines; example shown is for audiovisual 'm'). The total response is similar to the auditory-only format (dashed line). (**E**) For noisy speech, there is a reduction in the amplitude of the response to auditory phonemes for both preferred and non-preferred populations (example shown is for noisy auditory 'da'; only two neuronal populations are shown for simplicity). The signal-to-noise ratio (SNR) is defined as the ratio the response amplitude of the preferred to the non-preferred population. (**F**) For noisy audiovisual speech that is mouth-leading (example shown is for noisy audiovisual 'da') the response to the compatible neuronal populations are enhanced and the response to the incompatible neuronal populations are suppressed (visible as differences in bar height inside green and red outlines), resulting in increased SNR (red arrow).

DOI: https://doi.org/10.7554/eLife.48116.008

## Neuroanatomical substrates of the model

In the neural model, visual speech modulates the activity evoked by auditory speech. This is consistent with perception. For instance, silent viewing of the visual syllable 'ga' does not produce an auditory percept, but when paired with the auditory syllable 'ba', many participant report hearing 'da' (*Mallick et al., 2015*; *McGurk and MacDonald, 1976*). Neurally, this could reflect the fact that although the pSTG responds to visual-only speech, the response is much weaker than the response to auditory speech (note the different scales for the different modalities in *Figure 4A*). The weak response could be insufficient to create a conscious percept. Alternately, co-activation in early auditory areas together with pSTG could be required for an auditory percept. In either case, the significant correlation between the amplitude of the visual response and the degree of multisensory modulation bolsters the case that there is an interaction between visual and auditory speech information in pSTG.

In an earlier study, we demonstrated that audiovisual speech selectively enhances activity in regions of early visual cortex representing the mouth of the talker (*Ozker et al., 2018b*). In the present study, activity in auditory association cortex was differentially modulated by the presence or absence of a visual speech head start. These findings support the idea that interactions between auditory and visual speech can occur both relatively early in processing ('early integration') and at both early and late stages ('multistage integration') as proposed by *Peelle and Sommers (2015)*. Since cortex in superior temporal gyrus and sulcus receives inputs from earlier stages of the auditory and visual processing hierarchies, it seems probable that information about visual mouth movements arrives in pSTG from more posterior regions of lateral temporal cortex (*Bernstein et al., 2008*; *Zhu and Beauchamp, 2017*), while information about auditory phonemic content arrives in pSTG from posterior belt areas of auditory cortex (*Leaver and Rauschecker, 2016*).

## Testing the model

A simple neural model provides a qualitative explanation for the increased perceptual accuracy and decreased neural responses for mouth-leading compared with voice-leading words, many caveats are in order. The neural model assumes enhancement of compatible populations as well as suppression of incompatible populations. However, only suppression was observed in 25/28 pSTG electrodes (the remaining three electrodes showed similar responses to audiovisual and auditory-only speech). This observation is consistent with the fact that there are more incompatible than compatible phonemes for any particular viseme. If we assume that our relatively large recording electrodes recorded the summed responses of a combination of many suppressed populations and a few enhanced populations, the net effect should be suppression.

Using smaller recording electrodes permits recording from small populations of neurons selective for individual phonemes (*Hamilton et al., 2018*; *Mesgarani et al., 2014*). With these recordings, we would expect to observe both enhancement and suppression. For instance, an electrode selective

for the phoneme 'da' would be expected to show an enhanced audiovisual response (relative to auditory-only 'da') when the participant was presented with an auditory 'da' paired with the compatible visual open-mouth 'da' mouth shape, and a suppressed response when an auditory 'da' was paired with an incompatible mouth shape (such as 'f').

While the neural model provides an explanation for how enhancement and suppression could lead to improved perception of noisy speech, we did not directly test this explanation: only clear speech was presented in the neural recording experiments, and since the clear speech was understood nearly perfectly, it was not possible to correlate neural responses with perception. Therefore, a key test of the model would be to record the responses of phoneme-selective populations to noisy speech and correlate them with perception.

This would allow direct measurement of the signal-to-noise ratio (SNR) in pSTG by comparing the response of the neural populations responsive to the presented phoneme with the response of other populations. The model prediction is that the SNR in the pSTG should be greater for noisy audiovisual words than for noisy auditory-only words, and greater for mouth-leading words with a visual head start than voice-leading words without one.

In each participant, we recorded the response to only a few stimuli from a few electrodes, precluding meaningful analysis of classification accuracy. With a higher density of smaller electrodes, allowing recording from many pSTG sites in each participant, and a larger stimulus set, it should be possible to train a classifier and then test it on held-out data. The model prediction is that classification accuracy should increase for voice-leading words, due to the increased SNR generated by enhanced activity in neural populations representing the presented phoneme and decreased activity in neural populations representing other phonemes. On a trial-by-trial basis, the pSTG SNR would be expected to predict perceptual accuracy, with higher SNR resulting in more accurate perception. With large recording electrodes, the degree of suppression measured across populations should correlate with pSTG SNR (greater suppression resulting in greater SNR) and perceptual accuracy.

## Relationship to predictive coding and repetition suppression

Predictive coding is a well-established principle at all levels of the auditory system (*Denham and Winkler, 2017*; *Peelle and Sommers, 2015*). Cross-modal suppression may result from similar mechanisms as predictive coding, with the difference that the information about the expected auditory stimulus does not come from previously-presented auditory stimuli but from early-arriving visual speech information. This expectancy is generated from humans' developmental history of exposure to audiovisual speech, possibly through synaptic mechanisms such as spike-timing dependent plasticity (*David et al., 2009*). Over tens of thousands of pairings, circuits in the pSTG could be modified so that particular visual speech features inhibit or excite neuronal populations representing incompatible/compatible phoneme representations. The cross-modal suppression phenomenon we observed in pSTG may also be related to repetition suppression, in which repeated presentation of stimuli (in this case, successive visual and auditory presentations of the same phoneme) leads to sharpened representations and an overall reduction in response (*Grill-Spector et al., 2006*).

## The role of temporal attention

The simple neural model assumes that the visual speech head start provides an opportunity to rule in compatible auditory phonemes and rule out incompatible auditory phonemes in advance of the availability of auditory information from the voice. Another possibility is that the visual speech head start provides an alerting signal that auditory speech is expected soon, without being informative about the content of the auditory speech.

This is related to the idea of 'temporal attention' in which observers can use information about the timing of a sensory stimulus to improve detection and discrimination (*Warren et al., 2014*). One key test of the role of temporal attention in audiovisual speech perception is replacing the mouth movements of the talker with a simpler visual cue. In a recent study, Strand and colleagues found that a visually-presented circle that provided information about the onset, offset and acoustic amplitude envelope of the speech did not improve recognition of spoken sentences or words (*Strand et al., 2019*). In the present experiments, we did not present a non-face temporal attention cue, so we cannot disentangle the temporal and content cues provided by the mouth movements present for mouth-leading stimuli. In the neural recording experiments, there were only four stimulus

exemplars, so participants could have learned the relative timing of the auditory and visual speech for each individual stimulus, resulting in neural response differences due to temporal attention.

## Model predictions and summary

The neural model links the visual head start to an increased perceptual benefit of visual speech. In our experiments, voice-leading and mouth-leading words were different word tokens. Another approach would be to present the same tokens by experimentally manipulating auditory-visual asynchrony. Voice-leading speech could be transformed, advancing the visual portion of the recording and rendering it effectively 'mouth-leading' (*Magnotti et al., 2013*; *Sánchez-García et al., 2018*). Conversely, mouth-leading speech could be transformed by retarding the visual speech, rendering it effectively 'voice-leading'. The model predicts that the transformed voice-leading speech would not exhibit neural cross-modal suppression and the concomitant perceptual benefit, while transformed mouth-leading speech would exhibit both features.

Our findings contribute to the grown literature of studies showing how visual input can influence the auditory cortex, especially pSTG (*Besle et al., 2008*; *Ferraro et al., 2019*; *Kayser et al., 2008*; *Megevand et al., 2019*; *Zion Golumbic et al., 2013*). Together with previous work showing that visual cortex is modulated by auditory speech (*Ozker et al., 2018b*; *Schepers et al., 2015*), the present results from the pSTG provide another example of how cross-modal interactions are harnessed by all levels of the cortical processing hierarchy in the service of perception and cognition (*Ghazanfar and Schroeder, 2006*).

# Materials and methods

## Human subject statement

All experiments were approved by the Committee for the Protection of Human Subjects at Baylor College of Medicine.

## Overview of perceptual data collection and analysis

Perceptual data was collected using the on-line testing service Amazon Mechanical Turk (https://www.mturk.com/). Previously, we have shown that Mechanical Turk and in-laboratory testing produces similar results for experiments examining multisensory speech perception (*Magnotti et al., 2018*).

Within each participant, each stimulus exemplar was presented in four different formats: auditory-only (five trials); auditory-only with added auditory noise (10 trials); audiovisual (five trials); audiovisual with added auditory noise (10 trials). The trials were randomly ordered, except that in order to minimize carry over-effects all noisy stimuli were presented before any clear stimuli were presented. To construct the stimuli from the original audiovisual recordings, the auditory track of each exemplar was extracted using Matlab and all tracks were normalized to have similar mean amplitudes. To add auditory noise, white noise was generated with the same mean amplitude; the amplitude of the speech stimuli were reduced by 12 dB; the speech and noise auditory signals were combined; and the modified auditory track was paired with the visual track (for noisy audiovisual format). After each trial, participants responded to the prompt *'Type an answer in the text box to indicate what you heard. If you are not sure, take you best guess.'* No feedback was given.

The goal of the perceptual data analysis was to determine if the addition of visual information improved perception differently for mouth-leading and voice-leading words. Statistically, this was determined by testing the interaction (difference of differences) between word format (audiovisual *vs.* auditory-only) and word type (mouth-leading *vs.* voice-leading). While interactions can be tested with ANOVAs, accuracy data are proportional (bounded between 0% and 100%), violating the assumptions of the test. Instead, we applied a generalized linear mixed-effects model (GLMM) using odds-ratios (proportional change in accuracy, defined as the ratio of the probability of a correct response to the probability of an incorrect response) rather than absolute accuracy differences. For instance, an accuracy increase from 5% to 15% (absolute change of 10%) corresponds to a 3.4-fold increase in the odds-ratio (0.05/0.95: 0.15/0.85) while an accuracy increase from 50% to 60% (absolute change of 10%) corresponds to a 1.5-fold odd-ratio increase (0.5/0.5: 0.6/0.4).

A priori power was calculated using parameters from a previous study with similar methods (*Rennig et al., 2018*). Each participant was assigned a randomly-selected accuracy level for understanding auditory noisy speech, ranging from 0% to 50% across participants; adding visual speech increased accuracy within each participant by 30% for mouth-leading words and by 20% for voice-leading words. We sampled a binomial distribution using the actual experimental design of the first perceptual experiment, with 40 participants and 20 trials for each of the four conditions. The critical test was for the interaction within the GLMM between word type (mouth-leading and voice-leading) and format (auditory-only and audiovisual). In 10,000 boot-strapped replications, the power to detect the simulated 10% interaction effect was 80%.

Perceptual data and R code used for the data analysis and power calculations are available at https://openwetware.org/wiki/Beauchamp:DataSharing#Cross-modal_Suppression.

## Stimuli

The stimuli in the first perceptual experiment and the neural experiment consisted of two exemplars of mouth-leading speech ('drive' and 'last') and two exemplars of voice-leading speech ('meant' and 'known') presented in clear and noisy auditory and audiovisual formats (16 total stimuli). To estimate the auditory-visual asynchrony for these stimuli, Adobe Premier was used to analyze individual video frames (30 Hz frame rate) and the auditory speech envelope (44.1 kHz). Visual onset was designated as the first video frame containing a visible mouth movement related to speech production.

Auditory onset was designated as the first increase in the auditory envelope corresponding to the beginning of the speech sound. These values were: 'drive' 170 ms/230 ms (visual onset/auditory onset); 'last' 170 ms/270 ms; 'meant' 170 ms/130 ms; 'known' 200 ms/100 ms. In the second perceptual experiment, ten additional stimulus exemplars were presented in clear and noisy auditory-only and audiovisual format (40 total stimuli). The exemplars consisted of five mouth-leading words and five voice-leading words. The visual onset/auditory onset for the five mouth-leading words were: 'those' (125 ms/792 ms); 'chief' spoken by talker J (42 ms/792 ms); 'chief' spoken by talker L (167 ms/875 ms); 'hash' (42 ms/750 ms); 'vacuum' (42 ms/771 ms). The onsets for the five voice-leading words were: 'mature' (708 ms/625 ms); 'moth' (917 ms/833 ms); 'knock' (708 ms/458 ms) spoken by talker M; 'knock' (917 ms/813 ms) spoken by talker P; 'module' (833 ms/708 ms). The actual stimuli used in the recordings are available at https://openwetware.org/wiki/Beauchamp:Stimuli.

To visualize the complete time course of auditory and visual speech shown in *Figure 1*, two of the stimuli were re-recorded at a video frame rate of 240 Hz (these re-recorded stimuli were not used experimentally). The instantaneous mouth size was determined in each video frame using custom software written in R (*R Development Core Team, 2017*) that allowed for manual identification of 4 control points defining the bounding box of the talker's mouth. The area of this polygon was calculated in each frame and plotted over time. Auditory speech was quantified as the volume envelope over time, calculated by extracting the auditory portion of the recording, down-sampling to 240 Hz, and calculating the absolute value of the amplitude at each time step. The visual and auditory speech values were individually smoothed using a cubic spline curve with 30 degrees of freedom.

## Additional details of perceptual experiments

In the first perceptual experiment, 46 participants were presented with the first four stimulus exemplars (20 total stimuli) using Amazon Mechanical Turk (https://www.mturk.com/). Six participants had very low accuracy rates (from 0% to 75%) for clear words, suggesting that they were not attending to the stimuli. These participants were discarded, resulting in an *n* of 40 (adequate given the power calculation described above). In the second perceptual experiment, 46 MTurk participants were presented with the second set of 10 stimulus exemplars (40 total stimuli).

Because participant responses were collected using a text-box (free-response) preprocessing was required before analysis. We corrected basic spelling, typographical errors, well as mistaken plurals (e.g., drives was counted correct for drive); for the word 'last,' we counted 'lust' as an additional correct response because it was the single most frequent answer in both the clear and the noisy conditions. Analyses were conducted in R (*R Development Core Team, 2017*) using the *glmer* function (family set to binomial) from the *lme4* package (*Bates et al., 2015*). The initial GLMM contained fixed effects of word format (auditory-only *vs.* audiovisual) and word type (mouth-leading *vs.* voice-

leading), the word format-by-word type interaction, and a random effect for participant (different intercept for each participant) and exemplar. The baseline for the model was set to the response to mouth-leading words in the auditory-only word format. The inclusion of random effects allowed for participant differences but meant that the estimated odds-ratios are different than those calculated from the raw accuracy score.

For further analysis, separate GLMMs were created for each word type, with a fixed effect of word format (auditory-only *vs.* audiovisual), random effect of participant and baseline set to audiovisual word format. These separate GLMMs were used to calculate the reported odds-ratio differences and the significance within word type.

## Neural studies

Eight subjects (5F, mean age 36, 6L hemisphere) who were selected to undergo intracranial electrode grid placement for phase two epilepsy monitoring provided informed consent to participate in this research protocol. Electrode grids and strips were placed based on clinical criteria for epilepsy localization and resection guidance. The research protocol was approved by the Baylor College of Medicine Institutional Review Board. After a short recovery period following electrode implantation, patients were presented with audiovisual stimuli while resting in their hospital bed in the epilepsy monitoring unit. Stimuli were presented with an LCD monitor (Viewsonic VP150, 1024 × 768 pixels) placed 57 cm in front of the subject's face, and sound was projected through two speakers mounted on the wall behind and above the patient's head.

The four stimulus exemplars from the first perceptual experiment were presented in three formats: auditory-only, visual-only, and audiovisual (12 total stimuli). Auditory-only trials were created by replacing the speaker's face with a blank screen consisting only of a fixation target. Visual-only trials were created by removing the auditory component of the videos. No auditory noise was present in any format. Stimuli were presented in random interval. The behavioral task used a catch trial design. Subjects were instructed to respond only to an audiovisual video of the talker saying 'press'. No feedback was given. Neural data from catch trials was not analyzed.

## Neurophysiology recording and data preprocessing

Implanted electrodes consisted of platinum alloy discs embedded in flexible silastic sheets (Ad-Tech Corporation, Racine, WI). Electrodes with both 2.3 mm and 0.5 mm diameter exposed surfaces were implanted, but only electrodes with 2.3 mm were included in this analysis. After surgery, electrode tails were connected to a 128-channel Cerebus data acquisition system (Blackrock Microsystems, Salt Lake City, UT) and recorded during task performance. A reversed intracranial electrode facing the skull was used as a reference for recording, and all signals were amplified, filtered (high-pass 0.3 Hz first-order Butterworth, low pass 500 Hz fourth-order Butterworth), digitized at 2000 Hz, then converted from Blackrock format to MATLAB (MathWorks Inc, Natick, MA). Stimulus videos were presented using Psychtoolbox software package (*Brainard, 1997*; *Pelli, 1997*; *Kleiner et al., 2007*) for MATLAB. The auditory signal from the stimulus videos was recorded on a separate channel simultaneously and synchronously along with the electrocorticography voltage.

Preprocessing was performed using the R Analysis and Visualization of intracranial Electroencephalography (RAVE) package (https://openwetware.org/wiki/Beauchamp:RAVE). Data was notch filtered (60 Hz and harmonics), referenced to the average of all valid channels, and converted into frequency and phase domains using a wavelet transform. The number of cycles of the wavelet was increased as a function of frequency, from 3 cycles at 2 Hz to 20 cycles at 200 Hz, to optimize trade-off between temporal and frequency precision (*Cohen, 2014*). Data was down-sampled to 100 Hz after the wavelet transform. The continuous data was epoched into trials using the auditory speech onset of each stimulus as the reference ($t = 0$). For visual-only trials, $t = 0$ was considered to be the same time at which the auditory speech would have begun, as determined from the corresponding audiovisual stimulus.

## Calculation of broadband High-Frequency activity (BHA)

For each trial and frequency, the power data were transformed into percentage signal change from baseline, where baseline was set to the average power of the response from $-1.4$ to $-0.4$ s prior to auditory speech onset. This time window consisted of the inter-trial interval, during which

participants were shown a gray screen with a white fixation point. The percent signal change from this pre-stimulus baseline was then averaged over frequencies from 75 to 150 Hz to calculate the broadband high-frequency activity (BHA). Trials with median absolute differences more than five standard deviations from the mean were excluded (<2% excluded trials). For visualization in *Figure 3A*, the average BHA for auditory-only trials during auditory speech (time window 0.0 to 0.55 s) was calculated for each electrode and compared against baseline (single sample *t*-test). The resulting *p*-value was plotted according to a white to orange color scale (white is p=0.001, Bonferroni-corrected; orange is $p<10^{-14}$).

## Electrode localization and selection

FreeSurfer (*Dale et al., 1999*; *Fischl et al., 1999*) was used to construct cortical surface models for each subject from their preoperative structural T1 magnetic resonance image scans. Post-implantation CT brain scans, showing the location of the intracranial electrodes, were then aligned to the pre-operative structural MRI brain using the Analysis of Functional Neuroimaging (AFNI; *Cox, 1996*) package and electrode positions were marked manually on the structural surface model. Electrode locations were projected to the surface of the cortical model using AFNI. SUMA (*Argall et al., 2006*) was used to visualize cortical surface models with overlaid electrodes, and positions were confirmed using intraoperative photographs of the electrode grids overlaid on the brain when available. Cortical surface models with overlaid electrodes were mapped to the Colin N27 brain (*Holmes et al., 1998*) using AFNI/SUMA, allowing for visualization of electrodes from all subjects overlaid over one single brain atlas.

All analyses were performed on electrodes (*n* = 28 from eight participants; *Figure 3A*) that met both an anatomical criterion (located over the posterior superior temporal gyrus) and a functional criterion (significant BHA response to auditory-only speech). The anatomical border between anterior and posterior superior temporal gyrus was defined by extending a line inferiorly from the central sulcus to split the superior temporal gyrus into anterior and posterior portions. The functional criterion was a significant (p<0.001, Bonferroni-corrected) BHA response to the auditory-only word format during the period from stimulus onset to stimulus offset (0 s to 0.55 s). Because the functional criterion ignored word type (voice-leading *vs.* mouth-leading) and did not include audiovisual stimuli, the main comparison of interest (the interaction between response amplitude for different word types and stimulus word formats) was independent of the functional criterion and hence unbiased.

## Statistical analysis of neural data

Neural responses were collapsed into a single value by averaging the high-frequency activity (BHA) across the time window from stimulus onset to stimulus offset (0 s to 0.55 s) for each trial for each electrode. These values were then analyzed using a linear mixed-effects model (LME) with a baseline of mouth-leading words in the auditory-only word format. The model factors were two fixed effects of word format (auditory-only *vs.* audiovisual) and word type (mouth-leading *vs.* voice-leading), a word format-by-word type interaction was included, as well as random effects of electrode nested within subject (random intercepts and slopes for each subject-by-electrode pair). Because electrodes were selected based on the response to auditory-only speech ignoring word type (regardless of response to audiovisual or visual-only speech), the model was unbiased. All LMEs were performed in R using the *lmer* function in the *lme4* package. Estimated *p*-values were calculated using the Satterthwaite approximation provided by the *lmerTest* package (*Kuznetsova et al., 2017*).

To further explore the word format-by-word type interaction, we created separate LMEs for each word type (mouth-leading and voice-leading). For each word type, the LME had fixed effect of word format (auditory-only *vs.* audiovisual) and random effects of electrode nested within subject, with baseline set to the auditory-only word format. These separate LMEs were used to calculate the significance and magnitude of the effect of auditory-only and audiovisual word format on BHA in pSTG within word type.

For the analysis shown in *Figure 4*, the average BHA for each visual-only word format trial was calculated over the time window from −0.1 to 0.1 s, the time interval when visual speech was present for mouth-leading words but absent for voice-leading words. We then created an LME model with fixed effect of word type and random effects of electrode nested within subject. The neural response onset stim was measured by calculating the average time (across trials) it took the BHA to

reach half its maximum value for each word format and word type. Paired *t*-tests were used to compare half-maximum times between specific word types and word formats.

## Acknowledgements

This research was supported by NIH R01NS065395 and R25NS070694.

## Additional information

### Funding

| Funder | Grant reference number | Author |
|---|---|---|
| National Institute of Neurological Disorders and Stroke | R01NS065395 | Michael S Beauchamp |
| National Institute of Neurological Disorders and Stroke | R25NS070694 | Patrick J Karas<br>Daniel Yoshor |
| National Institute of Mental Health | R24MH117529 | Michael S Beauchamp |
| National Institute of Neurological Disorders and Stroke | U01NS098976 | Michael S Beauchamp |
| National Institute on Deafness and Other Communication Disorders | F30DC014911 | Lin L Zhu |

The funders had no role in study design, data collection and interpretation, or the decision to submit the work for publication.

### Author contributions

Patrick J Karas, Data curation, Software, Formal analysis, Visualization, Methodology, Writing—original draft, Writing—review and editing; John F Magnotti, Conceptualization, Data curation, Software, Formal analysis, Supervision, Validation, Visualization, Methodology, Writing—original draft, Writing—review and editing; Brian A Metzger, Validation, Writing—review and editing; Lin L Zhu, Formal analysis, Validation; Kristen B Smith, Online data collection from Amazon MTurk; Daniel Yoshor, Resources, Data curation, Supervision, Funding acquisition, Writing—review and editing; Michael S Beauchamp, Conceptualization, Resources, Supervision, Funding acquisition, Validation, Visualization, Writing—original draft, Project administration, Writing—review and editing

### Author ORCIDs

Patrick J Karas (iD) http://orcid.org/0000-0002-2605-8820
John F Magnotti (iD) https://orcid.org/0000-0003-2093-0603
Michael S Beauchamp (iD) https://orcid.org/0000-0002-7599-9934

### Ethics

Human subjects: Informed consent to perform experiments and publish data was obtained from all iEEG participants. Approval for all parts of this study was obtained through the Baylor College of Medicine Institutional Review Board, protocol numbers H-18112, H-9240 and H-36309.

### Decision letter and Author response

Decision letter https://doi.org/10.7554/eLife.48116.014
Author response https://doi.org/10.7554/eLife.48116.015

## Additional files

### Supplementary files

• Transparent reporting form

DOI: https://doi.org/10.7554/eLife.48116.009

## Data availability

The RAVE Software used for analysis is freely available at https://github.com/beauchamplab/rave. Data have been deposited in Dryad: https://doi.org/10.5061/dryad.v815n58.

The following dataset was generated:

| Author(s) | Year | Dataset title | Dataset URL | Database and Identifier |
|---|---|---|---|---|
| Karas PK, Magnotti JF, Metzger BA, Zhu LL, Smith KB, Yoshor D, Beauchamp MS | 2019 | Data from: The Visual Speech Head Start Improves Perception and Reduces Superior Temporal Cortex Responses to Auditory Speech | https://doi.org/10.5061/dryad.v815n58 | Dryad Digital Repository, 10.5061/dryad.v815n58 |

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
