## [Decision Letter]

Thank you for submitting your article "Cross-modal suppression of auditory association cortex by visual speech as a mechanism for audiovisual speech perception" for consideration by *eLife*. Your article has been reviewed by two peer reviewers, and the evaluation has been overseen by a Reviewing Editor and Andrew King as the Senior Editor. The following individual involved in review of your submission has agreed to reveal their identity: Bradford Mahon (Reviewer #1).

The reviewers have discussed the reviews with one another and the Reviewing Editor has drafted this decision to help you prepare a revised submission.

Summary:

The manuscript was well received by both reviewers who found the work important and informative and the paper well-written. However, they raised a number of issues that you are encouraged to address before the paper can be considered further for publication in *eLife*. These are listed below.

Essential revisions:

1) Please address the need for classification of responses to different speech sounds used in the study. Such classification would strengthen the conclusion that multisensory integration actually takes place in pSTG. The reviewers would also like to see more details of the stimuli used in the study.

2) Please clarify that the assumptions used in the model determine its results (see comments reviewer 1).

3) Please address the role of attention in your discussion of the results (see comments from reviewer 2).

4) In the Discussion, please address potential limitations associated with the use of a limited set of stimuli and the use of exclusively clear speech in the study. Also, please comment on early vs. later multisensory effects and cite the Peelle and Sommers, 2015 paper.

*Reviewer #1:*

This is an interesting and important follow up study on an earlier paper published in *eLife* from the same group. In the current study the authors compare words that are mouth leading versus voice leading (i.e. where visible movement of the mouth either precedes or follows the initiation of auditory signal). The authors found that pSTG shows a response to mouth leading words that is not present for voice leading words (among a number of supporting analyses). This is an interesting and solid contribution and I hope that the comments below are constructive.

1) The core claim of the paper is that visual information contributes to multisensory prediction, and that this occurs in pSTG. The authors note in the Discussion that they did not do any classification of the speech sounds, but it seems that is the analysis that would be required to really demonstrate that the multisensory integration is actually happening in pSTG (the neural model that is presented does not speak to classification, but just to changes in signal amplitude). For instance, if the visual information in mouth leading words allows for exclusion of 80% of phonemes, then this should be reflected in classification accuracy correct? E.g., comparing classification of phonemes for audio only (ground truth) to the window of time for mouth leading words prior to onset of auditory information.

To be clear – I think that the paper makes an important contribution already – it’s just that the evidentiary status of the claim that multisensory integration is occurring in pSTG would suggest a more direct test, or at least some discussion about the specific predictions made for a classification analysis by the theory. (I do think if the classification data could be included it would be better, but I don't see this as a condition sin qua non for publication).

2) The outcome of the neural model seems to be entirely driven by the assumptions used in constructing the model-this should just be clarified that it is a demonstration of principle of the assumptions (to distinguish this from a model where first principles that do not explicitly make assumptions about increases or decreases in signal as a function of modality 'gives rise' to the observed phenomena).

*Reviewer #2:*

This manuscript presents research aimed at examining differences in neural activity in auditory association cortex between audio and audiovisual speech as a function of the temporal lag between visual speech and audio speech. The authors hypothesize that differences between A and AV responses should be larger when visual speech leads audio speech. This hypothesis appears to be confirmed in their data which shows a suppression of high gamma activity in pSTG for AV words relative to A words when visual speech leads audio speech, but not when audio speech leads visual speech. The authors interpret this – quite reasonably – as evidence for an effect of visual speech on multisensory integration. They further support this interpretation by showing that the strength of AV vs. A suppression is related to the size of the neural response to visual only speech.

I thought this was a well written manuscript on an interesting topic. I thought the experimental paradigm was somewhat suboptimal, but I then thought that the subsequent analyses were of very good quality and the overall results were very compelling and were interpreted in a very reasonable way with an interesting discussion. I have a few relatively minor comments.

1) My main concern with the work would be that the experiment involved the use of only a very limited set of stimuli – two mouth-leading and two voice-leading words. Ultimately the results are compelling, but I wonder about how confident we can be that these results would generalize to a broader set of stimuli, including natural speech. For example, I wondered about what effects repeating these stimuli might have on how the subjects are paying attention or otherwise engaging with the stimuli. I mean I can imagine that as soon as the subject sees the mouth leading stimulus, they know what audio word is going to come next and then they pay a bit less attention resulting in smaller responses. They don't get as much forewarning for the voice leading stimuli, so they don't drop their attention. I would have thought that might be worth discussing and would also suggest that the authors are very clear about their stimuli in the main body of the manuscript.

2) Another limitation – that the authors acknowledge – was the use of just clear speech in the electrophysiology experiments. I guess it might be worth mentioning that a bit more in the Discussion for two reasons: i) it might be worth highlighting that one might expect to see larger suppressive multisensory effects in (somewhat) noisy speech (in the section on model predictions for example), and ii) it casts a slight doubt on the idea that what we are seeing is genuinely multisensory – I mean the V is not very behaviorally very helpful for clear speech. That said, I do appreciate the result in Figure 4B provides evidence for a multisensory effect.

3) I thought it might be worth mentioning the Peelle and Somers, 2015 review and perhaps speculating about whether the results we are seeing here might reflect early vs. later multisensory effects – or whether you think that's a useful framework at all?

---

## [Author Response]

Essential revisions:1) Please address the need for classification of responses to different speech sounds used in the study. Such classification would strengthen the conclusion that multisensory integration actually takes place in pSTG.

We now address classification analysis in detail (see response to reviewer 1, comment #1, below, for complete text).

The reviewers would also like to see more details of the stimuli used in the study.

We have edited the Materials and methods section to provide more detail and made the stimuli freely available at https://doi.org/10.5061/dryad.v815n58

2) Please clarify that the assumptions used in the model determine its results (see comments reviewer 1).

We agree completely and have deleted the conceptual model from the manuscript and completely changed our description of the neural model (see response to reviewer 1, comment #2, below, for complete details).

3) Please address the role of attention in your discussion of the results (see comments from reviewer 2).

We now incorporate a discussion of attention (see response to reviewer 2, comment #1, below, for complete details).

4) In the Discussion, please address potential limitations associated with the use of a limited set of stimuli and the use of exclusively clear speech in the study.

We have performed new experiments with a larger stimulus set (see reviewer 2, comment #1) and have added material about the use of clear speech (see reviewer 2, comment #2 for complete text).

Also, please comment on early vs. later multisensory effects and cite the Peelle and Sommers, 2015 paper.

We now comment on early vs. later multisensory effects and cite the Peelle and Sommers, 2015 paper (see reviewer 2, comment #3 for complete text).

Reviewer #1:[…] 1) The core claim of the paper is that visual information contributes to multisensory prediction, and that this occurs in pSTG. The authors note in the Discussion that they did not do any classification of the speech sounds, but it seems that is the analysis that would be required to really demonstrate that the multisensory integration is actually happening in pSTG (the neural model that is presented does not speak to classification, but just to changes in signal amplitude). For instance, if the visual information in mouth leading words allows for exclusion of 80% of phonemes, then this should be reflected in classification accuracy correct? E.g., comparing classification of phonemes for audio only (ground truth) to the window of time for mouth leading words prior to onset of auditory information.To be clear – I think that the paper makes an important contribution already – it’s just that the evidentiary status of the claim that multisensory integration is occurring in pSTG would suggest a more direct test, or at least some discussion about the specific predictions made for a classification analysis by the theory. (I do think if the classification data could be included it would be better, but I don't see this as a condition sin qua non for publication).

We agree with reviewer 1 that classification analysis would be an important test for the neural model presented in the paper. Our existing data does not allow for a classification analysis, so we now present what reviewer 1 suggests as a reasonable alternative, namely "discussion about the specific predictions made for a classification analysis by the theory". From the Discussion:

“While the neural model provides an explanation for how enhancement and suppression could lead to improved perception of noisy speech, we did not directly test this explanation: only clear speech was presented in the neural recording experiments, and since the clear speech was understood nearly perfectly, it was not possible to correlate neural responses with perception. […] With large recording electrodes, the degree of suppression measured across populations should correlate with pSTG SNR (greater suppression resulting in greater SNR) and perceptual accuracy.”

2) The outcome of the neural model seems to be entirely driven by the assumptions used in constructing the model-this should just be clarified that it is a demonstration of principle of the assumptions (to distinguish this from a model where first principles that do not explicitly make assumptions about increases or decreases in signal as a function of modality 'gives rise' to the observed phenomena).

We agree completely. We have deleted the "conceptual model" entirely from the manuscript, removed the reference to the neural model from the title of the manuscript and made clear in the Discussionthat the neural model is a post-hoc explanatory model (rather than one derived from first-principles). That said, the neural model makes a number of interesting predictions that are sure to spur further experiments so we feel it is a valuable part of the manuscript.

Reviewer #2:[…] 1) My main concern with the work would be that the experiment involved the use of only a very limited set of stimuli – two mouth-leading and two voice-leading words. Ultimately the results are compelling, but I wonder about how confident we can be that these results would generalize to a broader set of stimuli, including natural speech. For example, I wondered about what effects repeating these stimuli might have on how the subjects are paying attention or otherwise engaging with the stimuli. I mean I can imagine that as soon as the subject sees the mouth leading stimulus, they know what audio word is going to come next and then they pay a bit less attention resulting in smaller responses. They don't get as much forewarning for the voice leading stimuli, so they don't drop their attention. I would have thought that might be worth discussing and would also suggest that the authors are very clear about their stimuli in the main body of the manuscript.

We fully agree with this critique about our limited stimulus set. As suggested by the reviewer, in addition to describing the stimuli in Materials and methodswe now write in the main body of the manuscript (Results section):

"In the first perceptual experiment, 40 participants were presented with 16 word stimuli consisting of four stimulus exemplars (two mouth-leading words and two voice-leading words) in each of the four formats (clear auditory, noisy auditory, clear audiovisual, noisy audiovisual)."

and

"In contrast to the perceptual studies, where both clear and noisy speech was presented, in the neural experiments only clear speech was presented in order to maximize the size of the neural response. The stimulus exemplars consisted of the two mouth-leading words and the two voice-leading words used in the first perceptual experiment presented in auditory-only, visual-only, and audiovisual formats (twelve total stimuli)."

To help address this concern, we have performed a new experiment using additional stimuli:

"In the second perceptual experiment, 46 participants were presented with 40 word stimuli different than those used in the first perceptual experiment, consisting of 10 stimulus exemplars (five mouth-leading words and five mouth-leading words) presented in each of the four formats."

The results of this new experiment reproduce and extend our findings to a much larger stimulus set:

"For these mouth-leading words, viewing the face of the talker increased the intelligibility of noisy auditory speech by 53%…For voice-leading words, viewing the face of the talker provided only a 37% accuracy increase…The interaction between format and word type was significant (*p* < 10^-16^) driven by the larger benefit of visual speech for mouth-leading words."

The fact that our findings replicate in a different and larger sample is an important confirmation. However, it is true that we cannot rule out alternative explanation. We now include a new section in the Discussion:

“The Role of Temporal Attention

The simple neural model assumes that the visual speech head start provides an opportunity to rule in compatible auditory phonemes and rule out incompatible auditory phonemes in advance of the availability of auditory information from the voice. […] In the neural recording experiments, there were only four stimulus exemplars, so participants could have learned the relative timing of the auditory and visual speech for each individual stimulus, resulting in neural response differences due to temporal attention.”

2) Another limitation – that the authors acknowledge – was the use of just clear speech in the electrophysiology experiments. I guess it might be worth mentioning that a bit more in the Discussion for two reasons: i) it might be worth highlighting that one might expect to see larger suppressive multisensory effects in (somewhat) noisy speech (in the section on model predictions for example), and ii) it casts a slight doubt on the idea that what we are seeing is genuinely multisensory – I mean the V is not very behaviorally very helpful for clear speech. That said, I do appreciate the result in Figure 4B provides evidence for a multisensory effect.

We agree that this is a very important point. We have added a new Figure 5E and provide additional material about this in the Discussion:

"The post hocneural model provides a qualitative explanation for the decreasedneural response to words with a visual head start. […] This process is illustrated schematically for noisy auditory "da" and noisy audiovisual "da" in Figure 5E."

In a different section of the Discussion:

"While the neural model provides an explanation for how enhancement and suppression could lead to improved perception of noisy speech, we did not directly test this explanation: only clear speech was presented in the neural recording experiments, and since the clear speech was understood nearly perfectly, it was not possible to correlate neural responses with perception. […] The model prediction is that the SNR in the pSTG should be greater for noisy audiovisual words than for noisy auditory-only words, and greater for mouth-leading words with a visual head start than voice-leading words without one."

3) I thought it might be worth mentioning the Peelle and Sommers, 2015 review and perhaps speculating about whether the results we are seeing here might reflect early vs. later multisensory effects – or whether you think that's a useful framework at all?

We cite the Peele and Somers review and now write in the Discussion:

“In an earlier study, we demonstrated that audiovisual speech selectively enhances activity in regions of early visual cortex representing the mouth of the talker (Ozker et al., 2018b). […] Since cortex in superior temporal gyrus and sulcus receives inputs from earlier stages of the auditory and visual processing hierarchies, it seems probable that information about visual mouth movements arrives in pSTG from more posterior regions of lateral temporal cortex (Bernstein et al., 2008; Zhu and Beauchamp, 2017), while information about auditory phonemic content arrives in pSTG from posterior belt areas of auditory cortex (Leaver and Rauschecker, 2016).”